# Antibody reactions of horses against various domains of the EHV-1 receptor-binding protein gD1

**Andreina Schramm, Mathias Ackermann⊙, Catherine Eichwald⊙, Claudio Aguilar, Cornel Fraefel⊙, Julia Lechmann⊙***

Institute of Virology, Vetsuisse Faculty, University of Zurich, Zurich, Switzerland

* julia.lechmann@uzh.ch

**Data Availability Statement:** All relevant data are within the paper and its Supporting Information files.

## Abstract

Equid alphaherpesviruses 1 (EHV-1) and 4 (EHV-4) are closely related and both endemic in horses worldwide. Both viruses replicate in the upper respiratory tract, but EHV-1 may additionally lead to abortion and equine herpesvirus myeloencephalopathy (EHM). We focused on antibody responses in horses against the receptor-binding glycoprotein D of EHV-1 (gD1), which shares a 77% amino acid identity with its counterpart in EHV-4 (gD4). Both antigens give rise to cross-reacting antibodies, including neutralizing antibodies. However, immunity against EHV-4 is not considered protective against EHM. While a diagnostic ELISA to discriminate between EHV-1 and EHV-4 infections is available based on type-specific fragments of glycoprotein G (gG1 and gG4, respectively), the type-specific antibody reaction against gD1 has not yet been sufficiently addressed. Starting from the N-terminus of gD1, we developed luciferase immunoprecipitation system (LIPS) assays, using gD1-fragments of increasing size as antigens, i.e. gD1_83 (comprising the first 83 amino acids), gD1_160, gD1_180, and gD1_402 (the full-length molecule). These assays were then used to analyse panels of horse sera from Switzerland (n = 60) and Iceland (n = 50), the latter of which is considered EHV-1 free. We detected only one true negative horse serum from Iceland, whereas all other sera in both panels were seropositive for both gG4 (ELISA) and gD1 (LIPS against gD1_402). In contrast, seropositivity against gG1 was rather rare (35% Swiss sera; 14% Icelandic sera). Therefore, a high percentage of antibodies against gD1 could be attributed to cross-reaction and due to EHV-4 infections. In contrast, the gD1_83 fragment was able to identify sera with type-specific antibodies against gD1. Interestingly, those sera stemmed almost exclusively from vaccinated horses. Although it is uncertain that the N-terminal epitopes of gD1 addressed in this communication are linked to better protection, we suggest that in future vaccine developments, type-common antigens should be avoided, while a broad range of type-specific antigens should be favored.

**Funding:** The author(s) received no specific funding for this work.

**Competing interests:** The authors have declared that no competing interests exist.

## Introduction

Equid alphaherpesviruses 1 (EHV-1) and 4 (EHV-4) are both members of the *Alphaherpesvirinae* subfamily in the family of *Herpesviridae* [1, 2]. They are endemic in horse populations worldwide, with the exception of Iceland, which is considered free of EHV-1, while EHV-4 is highly prevalent [3–5]. Horses primarily get infected via aerosols, direct contact to infected horses or indirect contact via fomites [6–8]. Primary replication sites are the epithelial cells of the upper respiratory tract, typically resulting in mild respiratory signs such as fever and nasal discharge [6]. Whereas the EHV-4 infection remains local and is only rarely associated to severe complications, EHV-1 has the ability to establish a mononuclear cell-associated viraemia, thereby gaining access to its secondary replication sites, i.e. the endothelial cells in the pregnant uterus and the central nervous system [6–9]. Subsequently, a cascade of inflammatory response and thrombotic events ensues, eventually leading to tissue necrosis [10–12], abortion, and severe neurological illness, known as equine herpesvirus myeloencephalopathy (EHM) [8, 12, 13]. In spite of these biological and clinical differences, the two viruses are genetically and antigenically closely related, leading among others to the development of cross-reacting as well as cross-neutralizing antibodies [2, 6, 7].

As all herpesviruses, EHV-1 and EHV-4 establish lifelong latency, from which they can be reactivated, leading to excretion and transmission of infectious virus [6, 7, 12]. Although only one inactivated vaccine against EHV-1 is licensed in Switzerland, various inactivated, modified live (MLV) and recombinant vaccines have been developed and are used in several countries to prevent or attenuate severe infection courses, i.e. abortion and EHM [12, 14, 15]. Most of these vaccines are claimed to prevent respiratory disease and in some cases abortion, but none is certified for protection against EHM [17]. Both, older and more recent studies have shown that current vaccines are not able to significantly reduce EHV-1-viraemia, which is a prerequisite for the development of abortion and EHM [8, 14, 16].

At least two previous studies have shown that the receptor-binding protein, glycoprotein D (gD), drastically affects if not determines both the host range and the clinical severity of EHV-infections [17, 18]. Specifically, an EHV-1 in which the original gD (gD1) had been deleted and replaced by EHV-4 gD (gD4) lost its broad host range as well as its ability to cause neurological disease in horses [17, 18]. As EHV-4 infections, despite of raising neutralizing antibodies against EHV-1, only poorly protect against EHV-1-associated disease, we hypothesize that specific immune responses against type-specific epitopes of gD1 may be important for protection [7, 19]. In a first step, we therefore wanted to identify type 1-specific antibody epitopes within the gD1 amino acid sequence. Both gD1 and gD4 consist of 402 amino acids (aa), including a signal peptide of 35 (gD1) and 30 (gD4) aa, respectively. Moreover, a 23 aa long transmembrane domain anchors both molecules into the viral and cellular membranes, respectively. Also, both molecules comprise four N-glycosylation sites within their extracellular domains. The aa-identity level between the two molecules amounts to 77% but the longest stretch of consecutive identical aa maps to the extracellular domain near the transmembrane region and ranges from aspartic acid 261 (D261) to threonine 299 (T299) [20, 21]. Although individual differences scatter all over the molecule, the highest density of consecutive non-identical aa maps close to the signal sequence at the amino termini of the two molecules (see S1 Fig) [17]. The aim of this study was to identify fragments of gD1 which are bound by type 1-specific antibodies.

Luciferase immunoprecipitation system (LIPS) assays have two advantages over conventional ELISA tests: (1) the antigens can be harvested and used in their native, non-denatured state and (2) the range of animal species to be tested can easily be extended because the newly formed immune complexes are precipitated by protein A/G-coated beads, which bind a wide range of immunoglobulins from different species [22–24]. Accordingly, we established four

parallel LIPS assays, in each of which increasing sized fragments of gD1 were fused to a nano-Luciferase to be used as antigens for the detection of fragment specific antibodies in horses. The shortest antigen encompassed aa 1–83 of the gD1 molecule (gD1_83), the next comprised aa 1–160 (gD1_160), then aa 1–180 (gD1_180), while the positive control molecule consisted of all 402 aa of gD1 (gD1_402). As Iceland is considered to be free of EHV-1 and therefore presumably free of gD1-specific antibodies while EHV-4 is widely prevalent, a collection of sera from Icelandic horses was used as test population [3–5]. In contrast, a collection of sera from Swiss horses, which likely had been exposed to EHV-1 and/or EHV-4 and some of which had a history of vaccination, was used as population in which gD1-specific antibodies against EHV-1 may have arisen.

As expected, nearly all horse sera, both Icelandic and Swiss, exhibited antibodies against the full-length gD1_402, which may be predominantly attributed to cross-reactive antibodies from EHV-4 infections. Surprisingly, antibodies against gD1_83 were restricted to horses with a history of vaccination against EHV-1.

## Materials and methods

### Horse sera

In this study, 60 horse sera from Switzerland (No. 1–60), 50 horse sera from Iceland (No. 61–110) and one serum from Germany (PosC2) were analysed. The Swiss horse sera were obtained from a previous Master project conducted at the Vetsuisse Faculty of the University of Zurich [25]. They had been collected from ten different stables in Eastern Switzerland and included healthy horses (22 geldings, 38 mares), with (n = 20) or without (n = 40) history of repeated vaccination against EHV-1 and EHV-4. The details of vaccination are unknown, although in Switzerland only one single, inactivated vaccine (Duvaxyn EHV 1,4) is licensed for use against EHV-1. Most of the Swiss horses were leisure horses (n = 21), followed by school horses (n = 20), breeding horses (n = 15) and sport horses (n = 4). The age of the horses ranged from 1–32 years, with a median age of 12 years. The Icelandic samples were kindly provided by Dr. Vilhjálmur Svansson (Institute for Experimental Pathology, University of Iceland, Iceland) and included 47 male and 3 female horses. Since Iceland is considered EHV-1 free and vaccination is prohibited, sera from the Icelandic horses served as a pool, which was presumably free of native anti-EHV-1 antibodies [3–5]. The age ranged from 0–9 years, with a median at 5 years. Unfortunately, there was no available data on the health or usage of the horses. The serum from Germany was from a gelding with unknown age, vaccination status and usage of the horse.

### ELISA

All 110 serum samples, including the validation sera No. 29, No. 62, posC1 and posC2 were tested in the EHV-1/EHV-4 Ab ELISA kit from Svanovir (Svanova, Uppsala, Sweden) [26, 27]. The assay targets a highly variable region of the glycoprotein G (gG) envelope proteins of EHV-1 (gG1) and EHV-4 (gG4) [26, 27]. According to the supplier's instructions, samples with OD values of 0.2 or higher are to be considered positive. Samples with OD values below 0.1 are to be considered negative, while samples in between of OD 0.1 and 0.2 are to be considered inconclusive. All samples were tested in two separate runs, each in single wells following the manufacturer's instructions. The mean value of the two runs was used to determine the serological state of each serum.

### Construction of expression vectors

The EHV-1 gD (gD1) sequences used in this study were derived from the neuropathogenic strain Ab4 (EHV1_AY665713) whose complete genomic sequence had originally been

published by Telford et al. in 1992 [20]. All constructs started with the native gD1 signal sequence, extended into the gD1 with increasing length (83 aa, gD1_83; 160 aa, gD1_160; 180 aa, gD1_180) up to the maximum of 402 aa (gD1_402), and were supplemented with a C-terminal nanoLuc luciferase (nLuc) tag, followed by a V5-epitope tag and a 6xHis tag. The desired amino acid sequences were reverse-translated and codon-optimized for expression in COS cells. Codon optimization and DNA synthesis were done by GenScript Europe, who also cloned the synthetic fragments under control of the cytomegalovirus immediate early (CMV-IE) promoter into the pcDNA3.1(+) plasmid vector, which is equipped with the SV40 replication origin for high protein expression in COS-7 cells.

## Transient expression in COS-7 cells

The nucleic acid sequence of all constructs was determined and compared to the original virus sequence. After in vitro translation it was checked if both–the virus and the synthetic sequence—code for the same protein. All constructs were correctly produced. Plasmid DNA was extracted using the Qiagen Plasmid Maxi Kit (Qiagen, Hilden, Germany) according to the manufacturer's instructions. The DNA concentrations of the maxi preps were measured using the NanoDrop One (Thermo Fisher Scientific, Waltham, Massachusetts, USA). COS-7 cells (kindly provided by Ola Sabet, University of Zurich) were grown in $75cm^2$ tissue culture flasks (TPP AG, Trasadingen, Switzerland) with Dulbecco's Modified Eagle's Medium (DMEM; Gibco, Carlsbad, USA) supplemented with 10% fetal bovine serum (FBS) (BioConcept, Allschwil, Switzerland) and 1% of antibiotic-antimycotic solution (Gibco) at 37˚C and 5% $CO_2$. The day prior to transfection, 12 million cells were seeded in a $150cm^2$ tissue culture flask (TPP AG) and incubated for 24h at 37˚C and 5% $CO_2$. For transfection, 12µg of maxi prep DNA were mixed with 1.5mL Opti-MEM (Gibco) and 60µL Plus Reagent (Invitrogen, Waltham, Massachusetts, USA), and incubated for eight minutes at room temperature (RT). Additional 1.5mL Opti-MEM were mixed with 90µL Lipofectamine LTX (Invitrogen). The Lipofectamine-mix was added to the DNA and incubated for 30 minutes at RT. After the cells were washed with 12mL Opti-MEM (Gibco), the DNA-Lipofectamine-mix was immediately added, and the cells were incubated for 4–5 h at 37˚C and 5% $CO_2$. Subsequently, the cells were washed twice with 12mL Opti-MEM and then incubated with 20mL of fresh DMEM for another 48h at 37˚C and 5% $CO_2$. For harvesting, the cells were scraped into the medium and transferred to 15mL tubes (Sarstedt, Nümbrecht, Germany). The cell suspension was then centrifuged for 10 minutes at 7200x g (Multifuge 3 S-qR Heraeus, Thermo Fisher Scientific). The supernatant, comprising the secreted antigens, was transferred to chilled 1.5mL tubes, whereas the cell pellet was lysed either in an equal volume of 2x Lämmli buffer (for Western immunoblotting) or else (for LIPS assays) in 210µL of non-denaturing lysis buffer containing a protease inhibitor (Immunoprecipitation Kit, Abcam, Cambridge, UK), and transferred to fresh, chilled 1.5mL tubes. Both the cellular fraction and the supernatant were sonicated for two cycles at 60% amplitude, 5s on/5s off for 25s at 4˚C (Q800R3 Sonicator, QSonica Sonicators, Newton, CT, USA). Finally, the samples were centrifuged at 21'100x g for 10 minutes (Biofuge fresco 17, Heraeus, Thermo Fisher Scientific), and the supernatant was transferred to fresh, chilled 1.5mL tubes and stored at -20˚C.

## Western blot

Expression of all gD-fusion proteins was checked by Western blot. In detail, 30µL of cell lysate were mixed with 5µL Laemmli protein sample buffer (6x; BioRad, Hercules, CA, USA) and boiled for 5 minutes at 95˚C. A 10% SDS-polyacrylamide gel was loaded with 15µL of cell lysate and 10µL of a marker (Chameleon Duo marker, Li-Cor Biosciences, Lincoln, NE, USA),

followed by electrophoresis and blotting onto a nitrocellulose membrane (0.45μm NC, Amersham Protran and Whatman 3MM CHR Filterpaper, Cytiva, Marlborough, Massachusetts, USA) at 100V and 400mA for 1.5 h. Next, the membrane was blocked with 5% skimmed milk on a rocking shaker at RT for 60 minutes. After washing the membrane with PBS-T for 10 minutes, it was incubated overnight at 4˚C with anti-V5 antibody (1:5'000; Invitrogen). The following day, the membrane was washed three times for 10 minutes with PBS-T and then incubated with a secondary antibody (1:15'000, IRDye 680RD Goat anti-Mouse, Li-Cor Biosciences, Lincoln, U.S.A.) for 1 h at RT. The membrane was washed three times with PBS-T for 10 minutes and then analysed in the Odyssey XF Dual Mode Imaging System (Li-Cor Biosciences).

## Bioinformatic analysis

Protein structures were predicted using AlphaFold [28] in a Google Colab notebook (ColabFold) [29]. Protein models were visualized and analyzed using UCSF ChimeraX (v1.7) [30].

## Identification of proteins by mass spectrometry

Cellular extract expressing gD1(402)-nLuc-V5-$H_{6X}$ and media supernatant of cells expressing gD1(83)-nLuc-V5-$H_{6X}$ were migrated in duplicated in SDS-polyacrylamide gels (SDS-PAGE) as described previously [31]. The supernatant of gD1(83)-nLuc-V5-$H_{6X}$ was precipitated with four volumes of acetone [32] and resuspended in 25 μl of Laemmli sample buffer (8% SDS, 40% glycerol, 10% mercaptoethanol, 200 mM Tris pH 6.8, 0.4% bromophenol blue). After migration, one half of each gel was transferred to nitrocellulose 0.45 μm for 2 h at 200 mA. The membrane was incubated with mouse anti-V5 tag antibody (ab27671, Abcam) followed by incubation with secondary goat anti-mouse-IRdye 800CW (Li-COR) and imaged at Li-COR Odyssey M Imaging System. The second half of each gel was stained by Coomassie blue using Imperial protein stain (ThermoFisher Scientific) and imaged at Li-COR Odyssey M Imaging System. The two images were aligned by molecular weight ladder. The specific band of the immunoblot was then correlated with its molecular weight range in Coomassie blue stained gel for direct band excision. The bands were analysed by mass spectrometry for protein identification at the Functional Genomics Center Zurich (FGCZ) of University of Zurich and ETH Zurich. The data were examined using Scaffold 4.4.2.

## LIPS assay

The protocol for the LIPS assay was adapted from Springer Protocols from "Immuno proteomics–methods and protocols" [33]. The gD-fusion antigens were solubilized with non-denaturing buffer: gD1_402 as cellular fraction was tested in a 1:100 dilution, gD1_83, gD1_160 and gD1_180 as supernatant in 1:2 dilution or undiluted. Nano-Glo luciferase assay reagent (Promega, Madison, Wisconsin, USA) was prepared following the manufacturer's instructions and 10μL were dispensed into individual wells of a Nunc F96 MicroWell flat bottom, white polystyrene plate (Thermo Fisher Scientific). Ten microliters of pre-diluted gD1-fusion protein were added to each well, and luciferase activity (relative light units (RLU/1 sec) was measured within 5 minutes in a luminometer (MicroLumat Plus, LB 96 V, up-version 2.0; software WinGlow v. 1.25.000003, Berthold Technologies, Bad Wilbad, DE) for 1 sec. Horse sera were diluted 1:100 with non-denaturing buffer, to which proteinase inhibitor had been added according to the manufacturer's instructions (Immunoprecipitation Kit, Abcam). In a Nunc MicroWell' round bottom, 96-well polypropylene plate (Thermo Fisher Scientific) the volume corresponding to $10^7$ RLU of gD-fusion antigen and 10μL of pre-diluted horse serum were added and topped-up with non-denaturing buffer to a final volume of 100μL. Plates were

sealed with MicroAmp Optical Adhesive Film (Thermo Fisher Scientific) and incubated over-night on a plate shaker (300rpm) at 4˚C. The next day, protein A/G sepharose beads were washed three times with 1mL wash buffer each (Immunoprecipitation Kit, Abcam) and centri-fuged at 2000x g for 2 minutes at 4˚C (Biofuge fresco 17, Heraeus, Thermo Fisher Scientific); the supernatant was aspirated after each step. The beads were then suspended in wash buffer as 30% slurry, and 10μL thereof were added to each well. The plate was sealed as described above and incubated on a plate shaker for 1 h at 4˚C. The content of the wells was transferred to fresh tubes and washed three times with 500μL wash buffer, followed by centrifugation at 2000x g for 2 min at 4˚ and aspirating the supernatant in between the wash steps. Then, 10μL of the slurry were transferred to a Nunc F96 MicroWell flat bottom, white polystyrene plate (Thermo Fisher Scientific). The required amount of nano-Glo reagent (Promega) was pre-pared, and 10μL of nano-Glo were added to each well. The plates were gently shaken on a plate shaker (300rpm) and luciferase activity was measured as described above.

**Validation and control sera.** Based on the ELISA results, four sera were chosen for valida-tion of the LIPS assay. Three sera with high OD-values (optical density) against gG1 and gG4 were selected as positive controls: No.29 (OD-values 1.57 EHV-1; 2.67 EHV-4, respectively) was obtained from a healthy mare from Switzerland, 18 years old, with no history of vaccination against EHV. The serum posC1 (OD-values EHV-1 1.48; 0.99 EHV-4) was from a 10-year-old, healthy Swiss mare which had a history of vaccination against EHV-1 and EHV-4. The serum posC2 (OD-values 3.45 EHV-1; 2.82 EHV-4; respectively) was obtained from a 10 year old Ger-man gelding with was also regularly vaccinated against EHV-1 and EHV-4, who had been tested positive for EHV-1-shedding by PCR two weeks prior to serum sampling. Serum No. 62 (OD-values 0.00 EHV-1; 0.02 EHV-4, respectively), which reacted negatively in both gG-ELISA tests, was selected as negative control serum. It had been obtained from a four-month-old foal from Iceland, whose data about health or vaccination were not available. The validation sera were tested three times in a LIPS assay against all gD1_fragments to evaluate binding of the various gD1-fragments. A paired t-test was performed to compare the RLU of each antigen (Prism 10, Version 10.0.2, Graphpad); p-values $\leq$ 0.05 were considered statistically significant and marked with an asterisk. Based on these results a negative control was selected and the cut-off was deter-mined as its average plus three times standard deviation (AVE + 3*STD).

**Sample testing.** All 110 sera from Switzerland and Iceland were tested against all four gD1-fragments in three separate runs, each in triplicate. The raw average RLU values of these nine replicates were calculated for each serum. To address intra- and inter-assay variabilities, these 10 values were compared against the average reaction of the negative control serum throughout all tests (inter-assay) as well as against its average reactions on the same plate (intra-assay). Sera reacting above the negative value in each of these comparisons, were consid-ered "true positive". Those always reacting equally or below the cut-off, were considered "true negative". Those that floated sometimes above and other times below the cut-off value were counted as "intermediate". Eventually, the results were stratified using these criteria and Mann-Whitney tests (Prism 10, Version 10.0.2) were used to identify significant (p<0.05) dif-ferences between the strata. The same approach was also used to stratify our data according to anamnestic information, including age, sex, and vaccination status of the horses as well as their status in the gG1 ELISA.

## Results

### ELISA

All horse sera, including our validation controls, were tested in a commercial ELISA for anti-bodies against type-discriminating fragments of the viral glycoprotein G (gG1 and gG4,

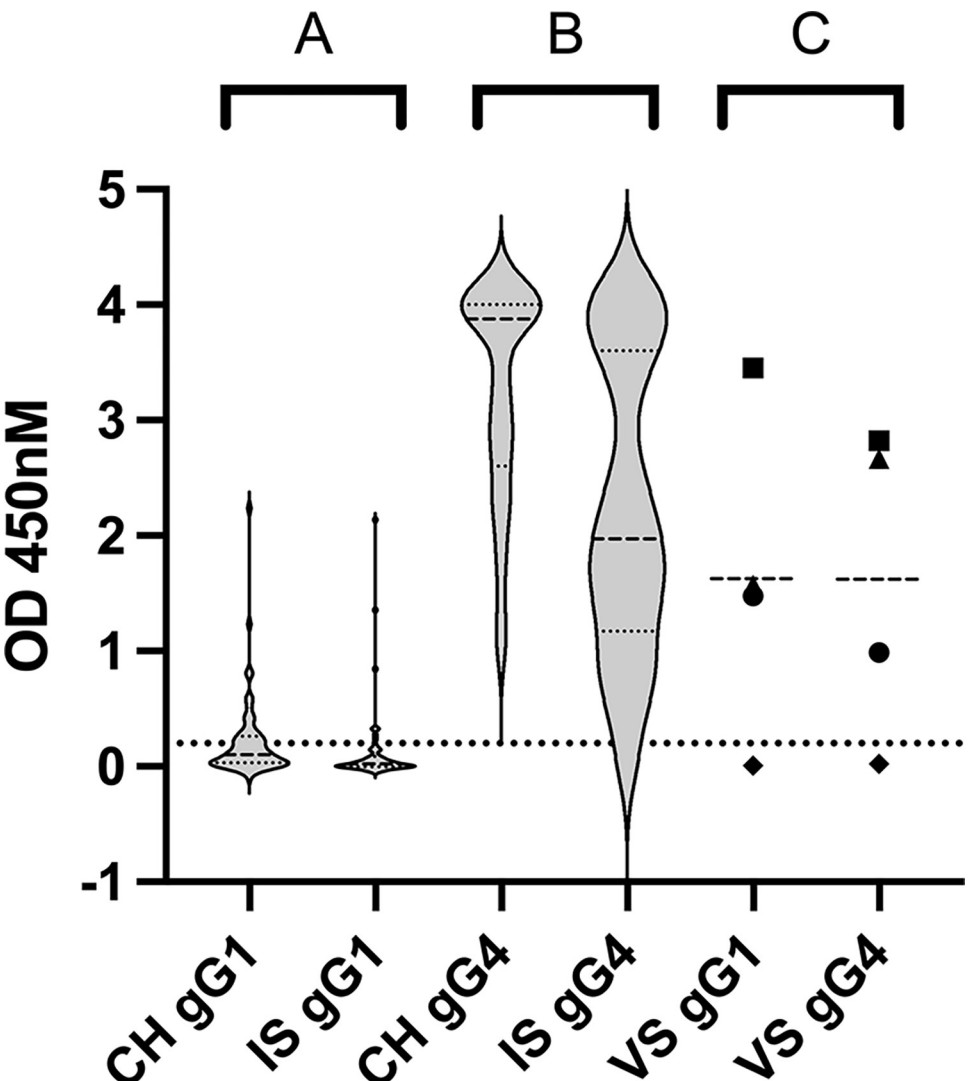

**Fig 1. Quantitative ELISA results of Swiss and Icelandic horse sera against gG1 and gG4, including validation sera.** (A) OD-values of Swiss and Icelandic horse sera against gG1. (B) OD-values of Swiss and Icelandic horse sera against gG4. (C) OD-values of validation sera: posC2 (■), posC1 (▲), No. 29 (●), and No. 62 (◆). CH = Swiss; IS = Icelandic; VS = validation sera; gG1 = glycoprotein G of EHV-1; gG4 = glycoprotein G of EHV-4; dotted line = cut-off at OD value of 0.2.

respectively). The ELISA results are shown in Figs 1 and 2. All samples from Swiss horses and all but two samples from the Icelandic horses contained detectable levels of antibodies against gG4. One of the Icelandic sera was deemed entirely negative, the second showed a weak reaction, which was deemed inconclusive. Positive reaction against gG1 was observed in 21 sera from the Swiss pool and 7 sera from the Icelandic; some sera reacted on the "inconclusive" level (Fig 2).

Therefore, all horses (except one) seemed to have a history of EHV-4 infection or vaccination, whereas a history of exposure to EHV-1 was less frequently confirmed on the serological level. One Icelandic horse serum (No. 62) (Fig 1 (VS = validation sera); VS gG1 and VS gG4) did neither react against gG1 nor against gG4 and was therefore selected as negative control for the consecutive validation of our LIPS assays. The sera No. 29, posC1 and posC2 (Fig 1) reacted with high OD-values against gG1 and gG4 and were selected as positive controls.

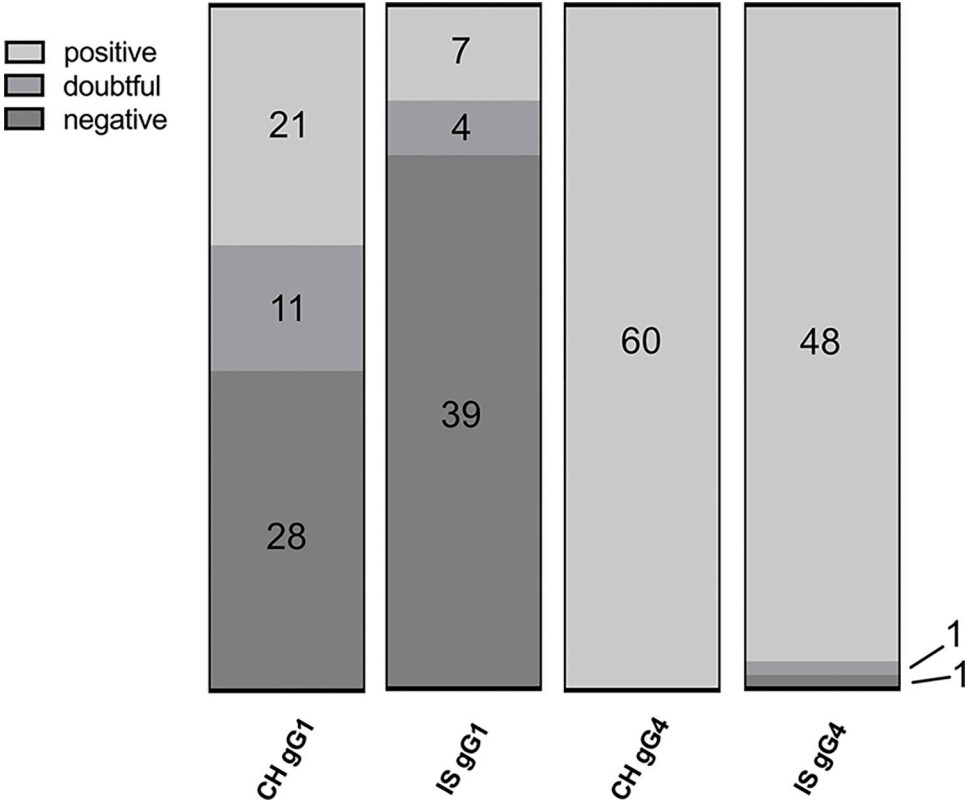

**Fig 2. Qualitative ELISA results (%) from Swiss and Icelandic horse sera against gG1 and gG4.** CH = Swiss;
IS = Icelandic; gG1 = glycoprotein G of EHV-1; gG4 glycoprotein G of EHV-4.

## Characterization of LIPS antigens

Separate dishes of COS-7 cells were transfected with each of the gD1 expression plasmids (see
Materials and Methods), while a mock-transfected dish served as negative control. After 48 h,
the cells and supernatants were harvested. Luciferase activity was determined in the soluble
contents of the supernatants before each of them was diluted to a starting luciferase activity of
$10^7$ RLU/50 µL. Next, a LIPS assay using an anti-V5 monoclonal antibody (mAb) was done
with each soluble antigen. As shown in Fig 3, the anti-V5 mAb precipitated 1–6 times $10^5$ RLU
of luciferase activity from each of the antigens. Notably, the gD1_83 fusion protein was most
efficiently precipitated, while the gD1_160 fragment provided the lowest values, although the
difference was less than 1 log, and the gD1_180 and gD1_402 reactions were in between.
When the same volume of the mock-transfected supernatant was used as control, luciferase
activity of less than 200 RLU was measured.

The cellular fractions were separated by PAGE, transferred to membranes, and analyzed by
Western immunoblotting using the same anti-V5 mAb. As shown in Fig 4, the mock-trans-
fected cells did not provide a band representing gD1. In contrast, multiple bands were
observed upon transfection of the gD1 expression plasmids. Two strong bands in the gD1_402
lysate at approx. 70 kDa corresponded well with the predicted 67 kDa non-glycosylated pre-
cursor of the full-length fusion protein and its glycosylated form, which is predicted to migrate
close to 70 kDa. In addition, several smaller and less intense bands were visible in the same
lane and may represent degradation products. The gD1_83 fusion protein yielded at double
band between 25 and 30 kDa, which also corresponded well with the predicted non-

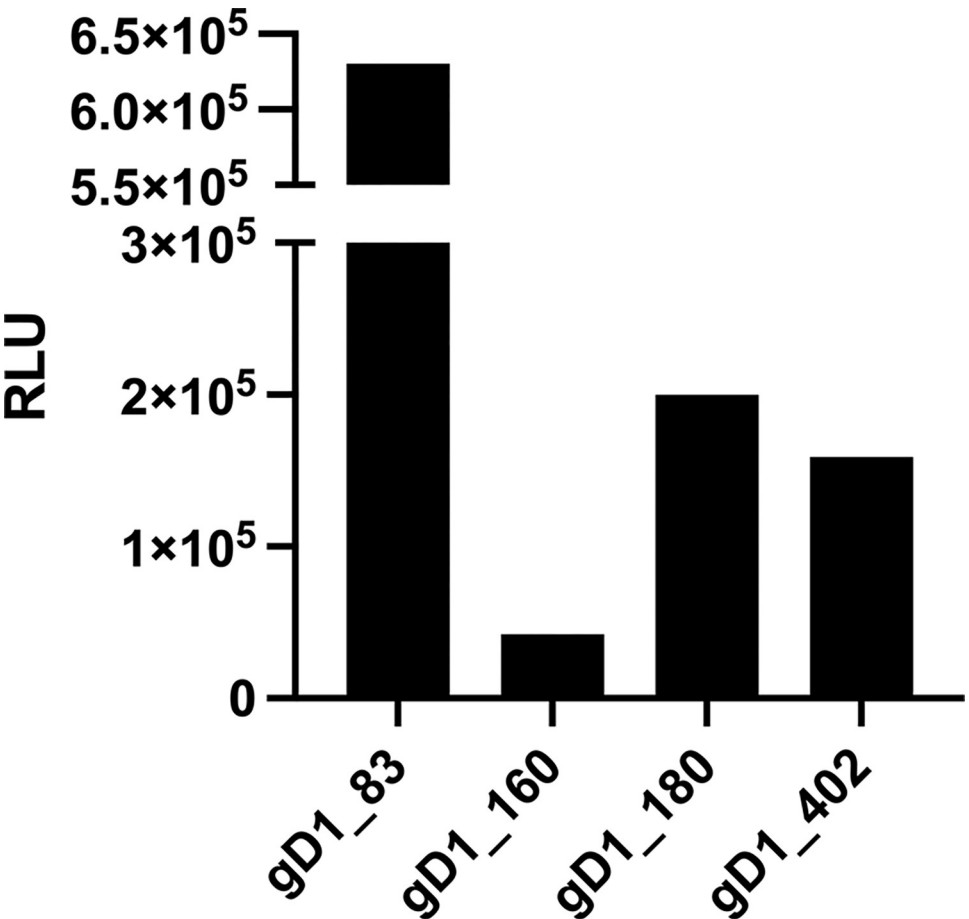

**Fig 3. Relative light units of LIPS antigens using anti-V5 mAb.** LIPS assay of the gD1_fragments gD1_83, gD1_160, gD1_180 and gD1_402 against the monoclonal anti-V5 mAb. Luciferase activity of mock antigen was less than 200 RLU and therefore not included in the graph. RLU = relative light unit.

glycosylated and glycosylated forms of this fusion protein. In the same sense, the gD1_160 and gD1_180 lysates formed double bands near the 38 kDa marker, consistent with the predicted MWs.

Having determined the apparent mobilities of the fusion proteins as well as the presence of the V5-tag and luciferase activity, we settled for mass spectrometry to formally give proof of the desired aa sequences, which represented the desired antigen, i.e. gD1 and its derivates, within these molecules. Indeed, the results showed that both gD1- and tag-sequences were present in the fusion proteins (S2 Fig).

Overall, we concluded from these observations that our LIPS antigens met the expected properties of identity and solubility for use in LIPS assays.

## LIPS assay

**Validation and controls.** In a first round of experiments, the selected control sera, hereafter termed validation sera, were assayed for their ability to precipitate the various LIPS antigens. As shown in Fig 5, the posC1 serum precipitated all four antigens at values of up to $10^5$ RLU, whereas the ELISA-negative serum No 62 precipitated more than two logarithmic scales less. The other two sera reacted at intermediate levels, showing the highest precipitation

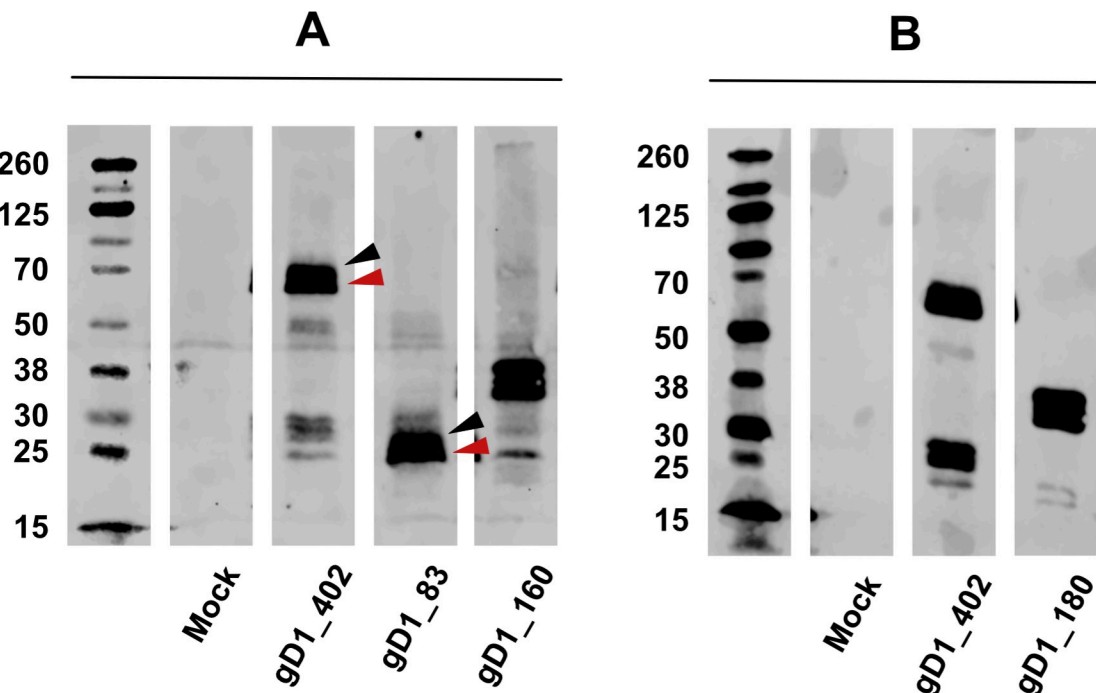

**Fig 4. Characterization of LIPS antigens by Western immunoblotting using anti-V5 mAb.** (A) Apparent mobility of gD1_402 (black arrowhead: glycosylated antigen 70 kDa, red arrow head: ungylcosylated antigen 67kDa), gD1_83 (black arrowhead: glycosylated antigen 30kDa, red arrow head: ungylcosylated antigen 25kDa) and gD1_160 (38 kDa). (B) Apparent mobility of gD1_180 (38kDa).

activity with the full-length gD1 fusion protein gD1_402. As expected, hardly any light emission (below 200 RLU) could be measured with extracts from mock-transfected cells. These latter data were therefore excluded from the graph.

Based on these observations and in combination with the previous ELISA data, posC1 and posC2 were selected as positive controls, whereas No. 62 was confirmed as negative control. Serum No. 29 was no longer used because of its low reaction against the smaller gD1 antigens.

Serum No. 62 with no apparent antibodies against EHV-1 and EHV-4 was important to determine the negative cut-off in our newly developed LIPS assays. Therefore, RLU of this negative control serum were determined as opposed to the positive controls in repetitive assays for at least 24 times (duplicates on 12 different plates). According to paired t-test, the results indicate that with all four antigen fragments, the negative control serum generated significantly (p-value <0.05) lower RLU values than the positive controls. However, using gD1_160 as antigen, the reactions of positive and negative control sera overlapped in most instances but was still statistically significant (p-value 0.0018).

In all four assays, the reactions of the negative serum with the different antigens clustered close to the bottom of the Y-axis (Fig 6). To establish negative cut-off values, the average values (AVE) and standard deviations (STD) of the negative serum were calculated and three STD were added to the AVE. Under these conditions, 23 of 24 values of the positive controls used against gD1_402 antigen exceeded the negative cut-off value, thus being deemed positive in the assay, which established a sensitivity of 95%. As none of the values from the negative control exceeded the same limit, the specificity of this test against the positive controls amounted to 100%. Similarly, 95% sensitivity and 100% specificity were calculated with gD1_180, whereas gD1_83 amounted to 100% sensitivity and 95% specificity. The low performing gD1_160 showed a sensitivity of just 25% but a specificity of 100%.

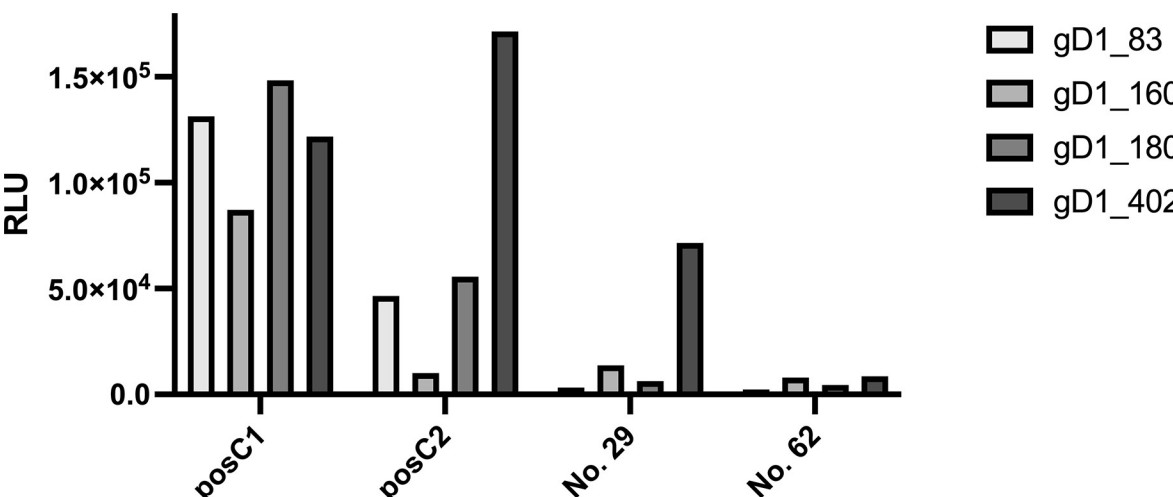

**Fig 5. Analysis of potential LIPS-control sera.** Each potential control serum was tested against each of the available antigens. Normally, extracts and supernatants from nLuc-transfected cells retained light emission values of about 20.000 RLU, the same materials from mock-transfected cells never emitted any light (RLU always below 200 RLU). Therefore, data obtained with mock antigens were not included in the graph. Grouped reactions of each antiserum against each antigen are along the x-axis, whereas the y-axis measures the corresponding RLUs. Bar-shading increases with length of the antigen.

We concluded from these experiments and calculations that three of our four newly developed LIPS assays (gD_83, gD_180 and gD_402) can consistently discriminate gD1-positive and -negative horse sera. In contrast, the fourth assay with gD1_160 antigen had a very low sensitivity.

**Sample testing.** Each of the horse sera in our collection was tested at least nine times against each antigen, i.e. as triplicates in three independent assays. The raw mean values of these nine replicates of each serum are presented in Fig 7. As the full-length gD1 molecule (gD1_402) comprises all cross-reacting epitopes shared with gD4 and, with one exception (negative control serum No. 62), all horses had tested positive in the EHV 4-ELISA, the reactions of all horse sera clustered well above the zero level. In contrast, the reactions against the smaller gD1 proteins clustered close to the zero level but with various numbers of individuals providing reactions well above zero. Notably, the reaction range of the Icelandic samples was in all assays much narrower than that of the Swiss samples.

While the values of positive reactions varied in between test replicates, the negative reactions–cut-off values as determined in the previous experiments–remained rather consistent.

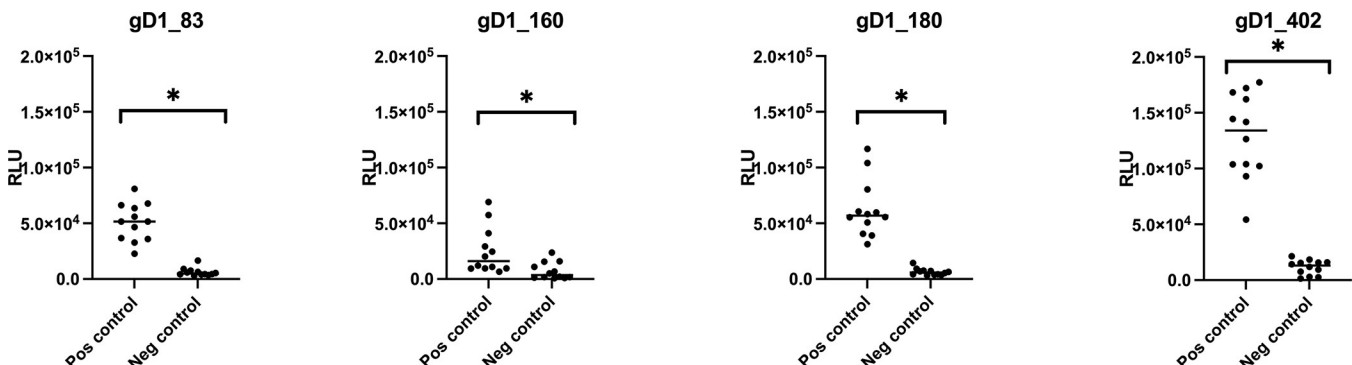

**Fig 6. Determining negative cut-off values.** All fragments generate significantly (p-value <0.05) lower RLU values for the negative control than the positive control. In fragment gD160 the values of the positive and the negative control were partially overlapping. Asterisk = p-value was significant <0.05.

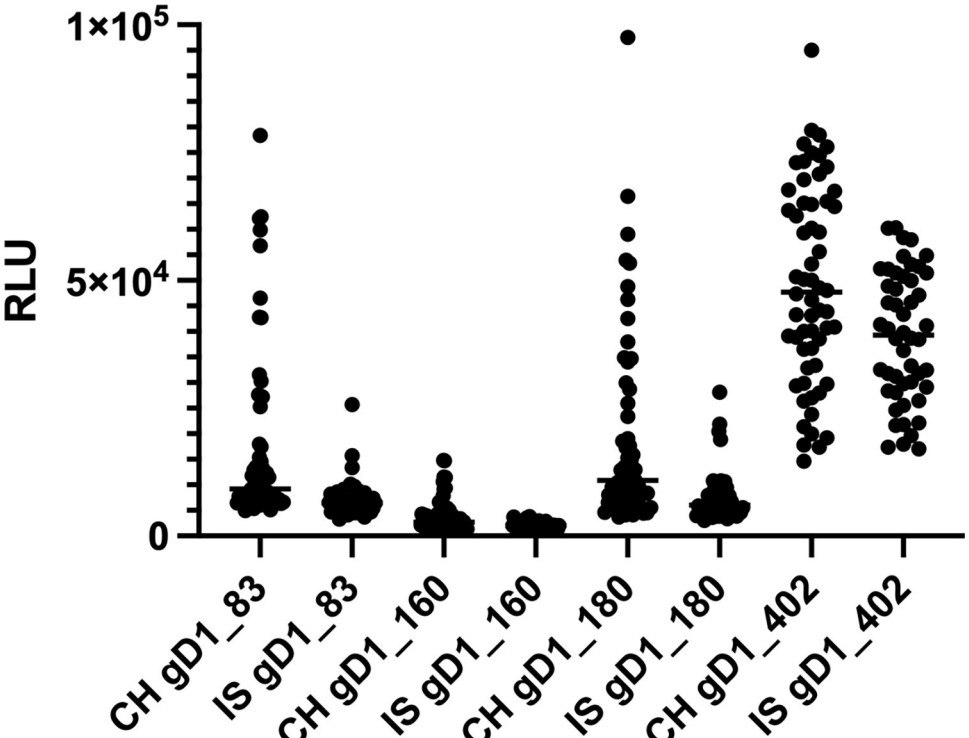

**Fig 7. Relative light units of horse sera tested against four different LIPS antigens.** Raw data of 60 Swiss and 50 Icelandic horse sera tested against four different gD1 fragments by LIPS assay. Each data point represents the mean of nine individual measurements of a serum.

This enabled us to stratify the sera into groups, the first of which comprised all sera that reacted consistently below the negative cut-off value calculated for each individual assay (negative, NEG). The second group comprised the sera which consistently reacted above these values (positive, POS), whereas the inconsistently reacting sera, i.e. those ranging sometimes above and other times below the cut-offs, were counted as intermediate group (intermediate, INT).

For Fig 8, the stratified median reactions of each individual serum against each antigen were plotted. Furthermore, Mann-Whitney tests, using all nine individual values per serum as well as their average, revealed that the POS and the NEG strata in the gD83 assay were significantly different (p <0.0001), thus, identifying the gD83-POS sera as "true positive" and the gD83-NEG sera as "true negative". In contrast, the NEG and the INT strata were not significantly different. Similarly, the gD180-POS and gD180-NEG strata were significantly different, whereas the gD160 assay was unable to discriminate between "true positive" and "true negative" individuals. Finally, no "true negative" individuals were identified in the gD402 assay, which was consistent with our previous ELISA results.

**AlphaFold analysis.** Noting that in contrast to the other three fragments, the gD1_160 fragment performed poorly in the above tests, the predicted 3D-structures of the four antigens were addressed by using the AlphaFold tool. In a first round, only the gD1-parts of the constructs were addressed in order to compare them against a known crystal structure as well as against the fusion proteins comprising the same gD1-fragments (Fig 9A–9D). Notably, the 3D structure predicted by AlphaFold was very similar to the published crystal structure of EHV1 gD [34]. Moreover, the structure did not noticeably change upon fusion with the C-terminal

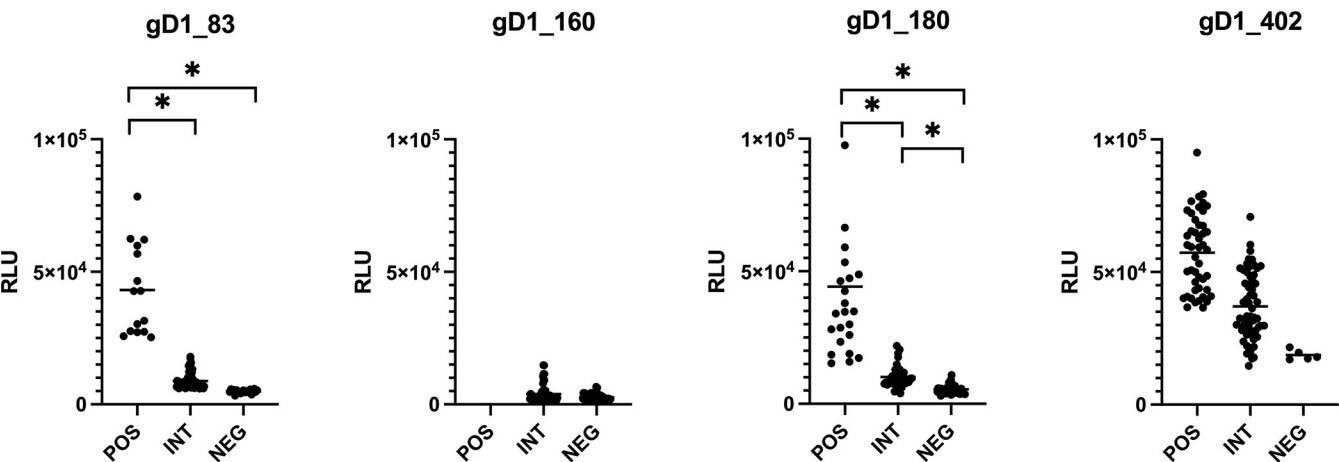

**Fig 8. Validation and stratification of horse sera reactions against various gD1 fragments.** Each horse serum was tested in three independent assays with each three replicates against four different gD1 antigens. Subsequently, the sera were stratified into three groups: "true positive" (POS), "intermediate" (INT, and "true negative" (NEG). The median values of these analyses are shown separately for each serum against each antigen. X-axis: stratification group; y-axis: relative light units (RLU) of the mean value of each serum. Bar: median of all sera in the group. Asterisks = p<0.05.

tail of the gD1_402 molecule, which consisted of the nLuc-V5-6xHis domains. In contrast, the gD1_83 antigen was predicted to remain unfolded, which also remained unaffected by the C-terminal tail (Fig 9A). The gD1_180 fragment was predicted to fold in a similar manner as the full-length glycoprotein, but its structure was even stabilized in the presence of the C-terminal tail (Fig 9H). The gD1_160 fragment, however, was predicted to assume a folding pattern that differed from the published crystal structure (Fig 9E–9G) as well as from the predicted structures of gD1_402 and gD1_180. Indeed, this observation provided a possible explanation for its poor performance in the LIPS assay (Fig 7).

**Comparison of anamnestic and serologic data.** While no anamnestic data were available for the Icelandic horse sera, various parameters were known for the Swiss horse sera. As the gD1_83 LIPS assay had been the only test to unanimously identify sera with "true" antibodies against gD1, the same data were used to stratify for sex, age, vaccination status, and gG1 antibody reaction with no statistically significant result (Fig 10). However, only two of the 14 "true positive" sera reacting with gD1_83 antigen came from unvaccinated horses, whereas 11 came from horses with documented history of repeated vaccination against EHV-1 and one serum came from a horse with unknown vaccination status. In contrast, only one of the vaccinated horses stratified among the "true negative" samples, whereas the remaining 6 sera stratified among the intermediate reactors. Surprisingly, only three of the gG1-positive horses stratified with vaccinated individuals.

## Discussion

In this study, a total of 110 horse sera sourced from both Switzerland (n = 60) and Iceland (n = 50) were screened in newly established luciferase immunoprecipitation system (LIPS) assays. The primary objective was to test the sera of horses with known exposure to EHV-4 and/or EHV-1 for antibodies targeting an N-terminal fragment of the receptor-binding glycoprotein D of EHV-1 (gD1). It is widely accepted that all horses worldwide are frequently exposed to EHV-4, a close relative to EHV-1 but with minor clinical importance [25, 35–38]. This difference is clearly associated with the viral receptor-binding gD molecules (gD1 and gD4, respectively), since replacement of the gD1 gene in the backbone of the EHV-1 genome with a gD4 gene negatively affected the viral host range and attenuated the clinical signs upon

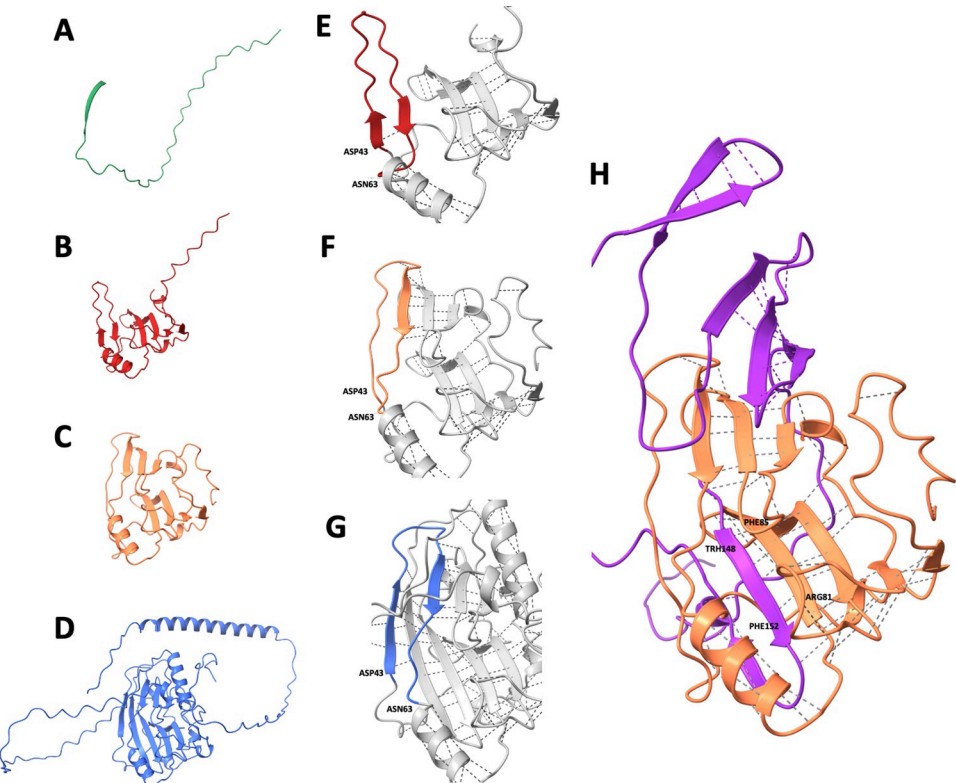

**Fig 9. Structure predictions and comparison of the gD1 fusion fragments.** Structure comparison of the different gD1 fragments when fused to nLuc-V5-6xHis. As the structures in A, B, and D remained unaffected by fusion to their nLuc-V5-6xHis tails, only the gD1 fragments are shown. **A** gD1_83, **B** gD1_160, **C** gD1_180, **D** gD1_402. The structure of the motif encompassing Asp43 to Asn63 had no effect on gD1_83 but forced a different structure on gD1_160 (magnified in E) as compared to the structures predicted for gD1_402 (magnified in G) and gD1_180 (F). As shown in panel H, the nLuc-V5-6xHis (purple) tail of gD1_180 (orange) had an effect on the molecule for increased similarity to the gD1_402 structure by stabilizing it via H-bonds between motifs Arg81-Phe85 (gD1_180) and Thr148-Phe152 (nLuc).

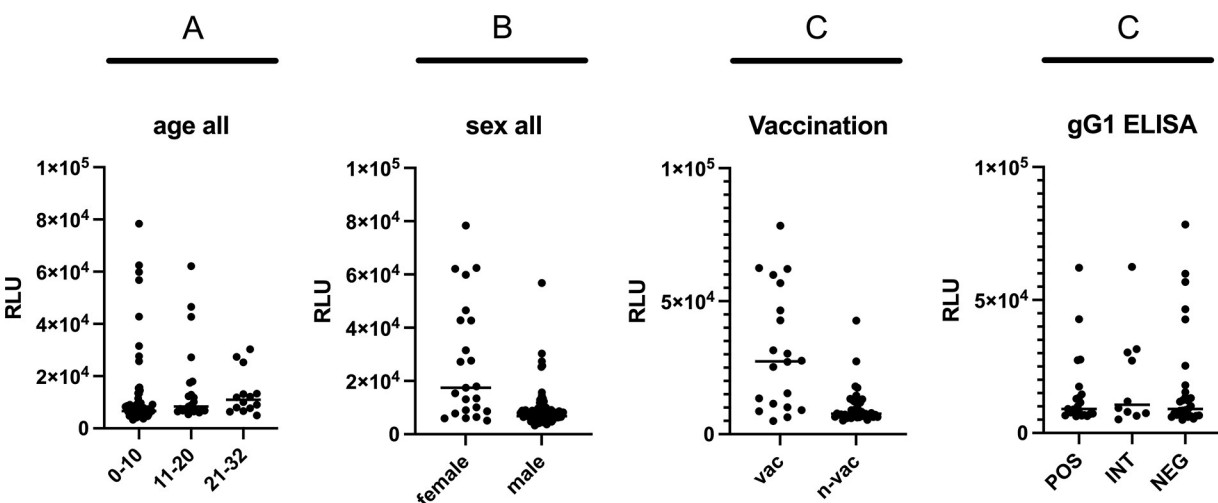

**Fig 10. Stratification of LIPS assay results using anamnestic data.** (A) Horse sera grouped in three different age categories (in years). (B) Horse sera differentiated by sex. (C) Horse sera differenciated by vaccination status (vac = vaccinated; n-vac = not vaccinted). (D) Horse sera differentiated by ELISA gG1 results. All tests were not statistically significant (p-value >0.05). RLU = relative light units.

infection of horses [17, 18]. Yet, the gD1 and gD4 molecules are still closely related, giving rise to cross-reacting neutralizing antibodies. On the amino acid level, the overall identity between gD1 and gD4 is 77% [20, 21]. Yet, many differences locate close to the molecule's N-terminus. The first 83 amino acids of gD1 (gD1_83) comprise the signal sequence (35 aa), which is supposed to be cleaved off by the signal peptidase [39] and not playing an important role as target for antibodies but is needed in the construct to send the native molecule to the secretory pathway. After cleavage, the remaining 48 aa fragment comprises two of the four conserved N-glycosylation sites, whose underlying amino acid sequences differ from the corresponding gD4 fragment. Moreover, only 29 of those 48 amino acids (60%) are identical between gD1 and gD4, thus, leaving 19 differing amino acids, spread out all over the fragment, which may offer epitopes for type-specific antibodies. With increasing length of the N-terminal fragment, the amino acid identity increases to 76% (gD_160) and 78% (gD1_180), respectively. While these three fragments are all thought to be secreted to the cell culture supernatant during transient expression in transfected cell cultures, the original molecule (gD1_402) will remain predominantly cell-associated, anchored through its transmembrane domain and cytoplasmic tail (see S3 Fig). Consequently, the full length gD1 molecule will be recognized by cross-reacting antibodies against gD1, which arise following exposure to either EHV-1 or EHV-4, as well as by type-specific antibodies, which emerge solely upon exposure to EHV-1 but not upon sole exposure to EHV-4. The smaller gD1 fragments carry a smaller number of antibody epitopes but the gD1_83 fragment is most likely to pick up antibodies that can only be observed following specific exposure to EHV-1. Indeed, these predictions could be confirmed throughout the initial validation of the new assays, upon using and evaluating the positive and negative control sera (see Fig 6). The same validation process revealed that the assays using gD1_402, gD1_180, and gD1_83 antigens were all capable of highly specifically and sensitively discriminating sera with or without antibodies against gD1 and/or its N-terminal fragments, whereas the assay using gD1_160 was not useful for that purpose. Among other possible factors (e.g. solubility, toxicity), this circumstance could be attributed to the different folding of gD_160 (Fig 9). During the validation process, it became obvious that positive sera showed quite variable values upon intra- and inter-assay comparisons. In contrast, the negative values were very stable and, therefore, used to determine cut-off values for each test and also to discriminate between "true positive" and "true negative" horse sera. Once these conditions and parameters were established, it was possible to stratify the results against anamnestic data.

Consistent with the current literature, the gG4-ELISA detected antibodies against EHV-4 in all but one of the horse sera [25, 35–38]. This single negative serum from an Icelandic horse was consequently used as negative control for all the newly developed assays (Fig 1). In contrast, the gG1 ELISA detected specific antibodies against EHV-1 in only 35% of the sera from the Swiss pool and in 14% of the sera from the Icelandic pool. Although these percentages fall within the reported seroprevalence of EHV-1 worldwide (ranging from 13–82%), the detection of seropositive Icelandic samples was unexpected, as Iceland is considered EHV-1 free and vaccination and import are prohibited (pers. comm. V. Svansson, Jan 18[th], 2024) [3–5, 25, 35–38].

Although with many borderline reactors, particularly with many sera from the Icelandic pool in the "intermediate" group (Figs 7 and 8), our newly developed gD1_402 assay mirrored the results of the gG4-ELISA, confirming the cross-reacting nature of neutralizing antibodies against EHV-1 and EHV-4, which are mainly directed against gD [2, 6, 7].

Whereas the gD1_160 assay was not useful, the other two antigen fragments, gD1_180 and gD1_83, were able to discriminate "true positive" and "true negative" samples. In particular, 13 out of 60 sera from Switzerland and one out of 50 from Iceland were identified as "true positive" in the gD1_83 assay (Fig 8). Thus, there were horses among our sampled populations that showed antibodies, which cannot be explained by exposure to EHV-4. The true positive

result in the Icelandic horse serum remains unexplained. The serum of this horse tested sero-negative for gG1 and seropositive for gG4 by ELISA. Although Iceland is considered EHV-1 free and vaccination is prohibited, there were also EHV-1 positive sera in the ELISA, the origin of which is unknown. However, upon stratification according to anamnestic information, it turned out that not a single one of those criteria was able to provide a significantly identifiable population (see Fig 10). Despite of interesting trends, "true positive" and "true negative" reactors were found among all age groups as well as among mares and geldings. It is well documented for many herpesviruses that seropositivity may remain for life, although younger individuals are more likely to have experienced recent exposures, resulting in higher antibody reactions [7, 36]. Under equal exposure conditions, mares and geldings have most likely the same probability to react positively or negatively in any given test for EHV antibodies. However, involvement in breeding may increase the risk of exposure for mares but, for obvious reasons, not as much for geldings. These trends can be seen in the corresponding figures (Fig 10), although they come without statistical significance.

Somewhat more surprising were the observations that not even the presence of antibodies against gG1 or history of repeated vaccination against EHV-1 stratified with the "true positives" against gD1_83. In this regard, three main considerations should be taken into account: (1) EHV-1 circulates less frequently and/or less efficiently than EHV-4, thus, providing a bias towards EHV-4-dependent and cross-reacting antibodies compared to EHV-1-specific antibodies. (2) The time after the last exposure certainly varies among the non-connected individuals, equally allowing the prevalence of high, intermediate or low titers of specific antibody titers against a particular EHV-antigen [40, 41]. (3) On the molecular level, gG1 and gD1 are independent antigens, thus, giving the individual ample opportunity to mount an antibody response primarily against this or that [2, 41].

The value of the above considerations became apparent, once the stratification of gD1_83-positive against all other criteria was more closely examined. The vaccinated horses were vastly over-represented in the gD1_83-positive group. Interestingly, only three gG1-positive individuals stratified among the vaccinated animals in the gD1_83-positive group. As mentioned above, this may be explained by the individual reaction against the two antigens, which may result in individual duration of antibody prevalence. Together, all these observations suggest that type-specific antibodies against EHV-1 can only rarely be identified, while cross-reacting antibodies common to EHV-1 and EHV-4 prevail very frequently. Whichever of those two virus types was responsible for the primary infection, will also generate memory cells for type-common epitopes. Thus, type-common memory cells will be present, even before the second type of EHV may arrive for a secondary, tertiary or n-tiary infection [42]. Considering that type-common antibodies are not protective against EHM, one may ask how helpful is it to include EHV-4 antigens into present vaccines [7, 19]. It is uncertain if the N-terminal epitopes of gD1 addressed in this communication are linked to better protection. However, our data confirm that type-specific antibodies against gD1 can be provoked by immunization. Therefore, we suggest that in future vaccine developments, type-common antigens should be avoided, while a broad range of type-specific antigens should be favored.

## Supporting information

**S1 Fig. Pairwise alignment of the gD1 and gD4 amino acid sequences.** Signal peptide (Signal) and transmembrane regions (TM) are indicated. Amino acids are counted from left to right, starting with the amino terminus of the molecule. Full bars in the Conservation graph (red bars) indicate aa identity; half bars indicate differences.
(TIFF)

**S2 Fig. Mass-spectrometry characterization for cell extract expressing gD1(402)-nLuc-V5-H6X and supernant of cells expressing gD1(83)-nLuc-V5-H6X. A)** Schematic representation of gD1(402) and gD(83) fused to nanoLuc, V5 tag, and histidine tag ($H_{6X}$), respectively. **B)** Coomassie blue staining (lanes 1 and 3) aligned with immunoblotting anti-V5 (lanes 2 and 4) of a cell extract of gD1(402)-nLuc-V5-$H_{6X}$ (left panel) and concentrated media supernatant of gD1(83)-nLuc-V5-$H_{6X}$ (right panel). **C)** Distribution of peptides coverage after MS analysis over the amino acid sequence of gD1(402)-nLuc-V5-H6X (upper panel) and gD1(83)-nLuc-V5-H6X (lower panel). The yellow highlighted amino acids correspond to the detected regions. **D)** Table indicating the probability (%), coverage peptide (%), and molecular weight for the indicated proteins. The green highlighted indicated 100% confidence in the result. (TIFF)

**S3 Fig. Characterization of LIPS antigens by Western immunoblotting using anti-V5 mAb.** Apparent mobility: gD1_402 70 kDA, gD1_83 25kDa and gD1_160 38 kDA. (JPG)

**S1 File. Raw images Western Blot.** (PDF)

**S2 File. Raw data ELISA and LIPS.** (XLSX)

## Acknowledgments

We would like to than Dr. Vilhjálmur Svansson (Institute for Experimental Pathology of the University of Iceland) and Prof. Dr. med.vet. PhD Angelika Schoster (Institute for Equine medicine of the Ludwig-Maximilian-University of München, Germany) for providing the sera for this study. Many thanks to Ola Sabet for providing us with COS-7 cells (Department of Molecular Life Sciences, University of Zurich).

## Author Contributions

**Conceptualization:** Mathias Ackermann.

**Data curation:** Andreina Schramm, Mathias Ackermann, Catherine Eichwald, Julia Lechmann.

**Formal analysis:** Andreina Schramm, Mathias Ackermann.

**Investigation:** Andreina Schramm, Catherine Eichwald, Claudio Aguilar.

**Methodology:** Andreina Schramm, Mathias Ackermann, Julia Lechmann.

**Project administration:** Cornel Fraefel.

**Resources:** Cornel Fraefel, Julia Lechmann.

**Supervision:** Mathias Ackermann, Cornel Fraefel, Julia Lechmann.

**Validation:** Andreina Schramm.

**Visualization:** Andreina Schramm, Mathias Ackermann, Catherine Eichwald, Claudio Aguilar.

**Writing – original draft:** Andreina Schramm, Mathias Ackermann, Julia Lechmann.

**Writing – review & editing:** Andreina Schramm, Mathias Ackermann, Catherine Eichwald, Claudio Aguilar, Cornel Fraefel, Julia Lechmann.

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
