## [Decision Letter · Decision Letter 0]

8 Jan 2024

PONE-D-23-35794Antibody reactions of horses against various domains of the EHV-1 receptor-binding protein gD1PLOS ONE

Dear Dr. Lechmann,

Thank you for submitting your manuscript to PLOS ONE. After careful consideration, we feel that it has merit but does not fully meet PLOS ONE’s publication criteria as it currently stands. Therefore, we invite you to submit a revised version of the manuscript that addresses the points raised during the review process.

We look forward to receiving your revised manuscript.

Kind regards,

Gianmarco Ferrara, PhD, MVD

Academic Editor

PLOS ONE

Reviewers' comments:

Reviewer's Responses to Questions

**Comments to the Author**

1. Is the manuscript technically sound, and do the data support the conclusions?

Reviewer #1: Yes

Reviewer #2: Yes

2. Has the statistical analysis been performed appropriately and rigorously? 

Reviewer #1: Yes

Reviewer #2: Yes

3. Have the authors made all data underlying the findings in their manuscript fully available?

Reviewer #1: Yes

Reviewer #2: Yes

4. Is the manuscript presented in an intelligible fashion and written in standard English?

Reviewer #1: Yes

Reviewer #2: Yes

5. Review Comments to the Author

Reviewer #1: The work aims to identify regions of gD1 that give rise to type-specific antibodies after infection with EHV-1 and/or vaccination. Cross reactive antibodies against gD1 and gD4 are known but knowledge about type-specific gD1 antibodies is limited. For that purpose horse sera from Swizerland and from Iceland, which is considered EHV1-free, were used in a diagnostic ELISA and adapted Luciferase immunoprecipitation system (LIPS) assays. Three antigens consisting of N-terminal gD1 protein fragments plus the full length protein were tested. Full length gD1 LIPS assay detected cross-reactive antibodies, however, gD1_83 fragment in LIPS assay seemed to identify gD1 specific antibodies in horse sera.

This work gives new insights into gD1 regions that can possibly give rise to type-specific antibodies which is important to understand the immune reaction after EHV-1 infection and vaccination. The results might help to improve vaccines which to date are not able to block an infection with either EHV-1 or EHV-4.

The data acquisition, assay validation, and statistical analysis seem sound. Weaknesses can be found in the interpretation of the data, placement into context of previous studies, and occasional spelling mistakes. I recommend the publication of this manuscript in PLOS ONE after revision.

Minor points

- line 57: “ EHV-1 in which the original gD (gD1) had been deleted and replaced by EHV-4 gD (gD4)” I am not sure that this statement of a deletion and replacement of gD is correct. Usually essential genes are not easily deleted in viruses. Please check and provide a reference.

- line 60-61: On what data do you build your hypothesis that type-specific epitopes of gD1 could be important for protection against infection?

- line 70-71: Unfortunately the sentence is unclear and with that the aim. There is probably a typo in the beginning of line 71. I understand the sentence in a way that the aim is to identify gD1 fragments that bind type-specific antibodies that arise after infection. If that is so, I wonder why mostly sera from horses was used that were vaccinated as stated in line 464-465 even more so since it is also interesting to see if specific antibodies are formed after vaccination?

- line 128, 131: You should choose one notation for the Cos cells. “Cos cells” or “COS-7 cells”.

- line 151: You probably meant “comprising” instead of “compromising”.

- line 159: I am a bit confused about this sentence. Was the amino acid sequence of the proteins checked or the nucleic acid sequence of the transfected DNA and how?

- line 254: There is an empty box next to the black filled square which probably should not be there.

- line 280-283: Can you please comment on what influence glycosylation of the protein fragments could have on your assay. Would this influence the antibody binding? And if so, would this have any effect on your LIPS assay?

- line 284-285: Since no gD1 specific antibodies were used in the western blot it might be helpful to verify the fragments by mass spectrometry.

- line 291-293: The figure legend should include the notion of glycosylated gD.

- line 411-412: You state that the gD1 fragments are thought to be secreted into the cell culture supernatant. Can you provide data for this claim? If I understood it right you used only cell lysate for western blotting where all protein fragments were apparently detected. It should be more clear if the presented data was generated with protein from cell lysate or supernatant.

- line 414-417: It is not clear to me why it is a consequence from protein being secreted into cell culture supernatant or not in an artificial system that only full length gD1 will be recognized by cross-reactive antibodies. Can you please comment on that and provide references?

- line 417-420: Is the prediction stated here based on any other study or data? Why would the shortest fragment be the one which is most likely to carry the type-specific epitope? Is this based on structural data?

- line 423-425: Do you have any idea why the gD1_160 did not work so well in your LIPS assay and why the positive sera values varied so much?

- line 442: How do you explain the one “true positive” result from an icelandic horse serum?

- line 458-462: For point 2 and 3 a references are needed. It would be helpful for the reader to have info about the kind of vaccines that are in use and the immune responses that are to be expected.

- line 473: A reference is needed for the statement that type-common antibodies are not protective against EHM.

- line 474: “It is uncertain that...” should be replaced by “It is uncertain if...”.

- Figure 7 and Figure 9 are the same.

- It would be really interesting to try to link the data to the structure of gD1 and gD4 protein in the discussion. It should be taken into account that fragmenting of proteins can result in different molecular structures which can influence antibody binding. A quick modeling via AlphaFold might help to see, if the structure of the gD1 fragments can be expected to correspond to full gD1.

Reviewer #2: Overall:

In this paper the authors investigate antibody responses to the related equine herpesviruses (EHV1 & 4, which exhibit significant sequence similarity). Former can give rise to EHM. What disease/clinical signs does EHV4 elicit?

Serum antibodies were detected via immunoprecipitation assay, to the gD glycoprotein (GP) via a series of N terminal fragments of increasing length. Compared with ‘type-specific’ ELISA. Horse sera from two distant and distinct countries (Switzerland and and Iceland) with different EHV incidence/vaccination.

Generally very well written, scientifically thorough and clear article.

Mostly very good, detailed description of M&M and Results.

Discussion raises some interesting points, some covered in a fulsome manner but other statements are not always followed up as fully as they could be – see recommendations in Minor Comments.

[Clarify if any EHV vaccination is carried out on Iceland – I presume not, as would impact study. I note brief mention in the M&M.]

Other comments which need addressing:

Abstract

Need to state Iceland was chosen as believed to be free of EHV1.

Either Luciferase should be in lower case or immunoprecipitation and system in uppers case (for the LIPS abbreviation) and used throughout.

Introduction

Line 17 – insert ‘respectively’ after gG4 in (gG1 and gG4)

Line 50 – suggest noted that these alternative vaccines used outside Switzerland

Line 68 – the region of non-identical amino acids should also be defined

Line 75 – briefly explain A/G-coated bead system

Line 85 – not that this would be due to cross-reactive antibodies

Materials & Methods

Line 128 – give source and variant # of COS cells and how these were codon optimized (as they are not human cells)

Line 129 – define cmv (also should be upper case - CMV)

Line 148 – ‘scrapped’ should read ‘scraped’.

Line 158/159 – sequencing line should be clarified – maybe ‘coding region’ rather than protein

Line 169 – ‘skimmed milk’

Line 217 – mention Graphpad software company

Results:

Line 245 – It is stated that 21 Swiss and 7 Icelandic sera were gG1 positive (a type-specific ELISA test), however it is then stated that ‘a history of EHV1 was only rarely confirmed’ – this is hardly ‘rare’. Rephrase – lower levels than EHV4 detection?

Line 263-267 are methods rather than results, so should be moved to M&M section.

Line 268 – should read 1-6x

Line 270 – the Fig 3 data for gD1_180 and 402 should be mentioned (for discussion later) as they are interestingly higher than for 160 (noting 160 also gave lower sensitivity result in Fig 5, as mentioned in later Results).

Line 344 – clarify ‘with one exception’.

Line 345 – rephrase ‘clouded’.

Discussion:

Line 403 – whole sentence needs clarifying – from production to transport to secretion (during infection or plasmid transfection or both?). What is cleaved off (and by which protease?). Is the 48aa truly a fragment or a section?

Line 417 – have any antibody epitopes been precisely identified? (e.g. mutation studies?)

Line 423 – authors should discuss why the gD_160 should give such poor responses and yet even 180 gives a reasonable response.

Line 435 – Authors should discuss this discrepancy – were different test assays used? Is monitoring often conducted. Do horses travel internationally from Iceland to events/breeding?

Line 462 – explain why you might see a preference in antibody response against the different antigens in different animals – is this referring to differences between vaccination and natural infection?

Line 472 – are the authors referring to ‘antigenic sin’? If so, how might this impact type-specific vaccination as suggested at the end of the Discussion?

6. PLOS authors have the option to publish the peer review history of their article (what does this mean?). If published, this will include your full peer review and any attached files.

Reviewer #1: **Yes: **Viviane Kremling

Reviewer #2: No

---

## [Author Response · Author response to Decision Letter 0]

28 Feb 2024

Dear reviewer, dear editor,

thank you for your valuable input. All our comments are included in the rebuttal letter.

Best regards

Julia Lechmann

---

## [Decision Letter · Decision Letter 1]

27 Mar 2024

Antibody reactions of horses against various domains of the EHV-1 receptor-binding protein gD1

PONE-D-23-35794R1

Dear Dr. Lechmann,

We’re pleased to inform you that your manuscript has been judged scientifically suitable for publication and will be formally accepted for publication once it meets all outstanding technical requirements.

Kind regards,

Gianmarco Ferrara, PhD, MVD

Academic Editor

PLOS ONE

Additional Editor Comments (optional):

Reviewers' comments:

Reviewer's Responses to Questions

**Comments to the Author**

1. If the authors have adequately addressed your comments raised in a previous round of review and you feel that this manuscript is now acceptable for publication, you may indicate that here to bypass the “Comments to the Author” section, enter your conflict of interest statement in the “Confidential to Editor” section, and submit your "Accept" recommendation.

Reviewer #1: All comments have been addressed

2. Is the manuscript technically sound, and do the data support the conclusions?

Reviewer #1: Yes

3. Has the statistical analysis been performed appropriately and rigorously? 

Reviewer #1: Yes

4. Have the authors made all data underlying the findings in their manuscript fully available?

Reviewer #1: Yes

5. Is the manuscript presented in an intelligible fashion and written in standard English?

Reviewer #1: Yes

6. Review Comments to the Author

Reviewer #1: (No Response)

7. PLOS authors have the option to publish the peer review history of their article (what does this mean?). If published, this will include your full peer review and any attached files.

Reviewer #1: **Yes: **Viviane Kremling

---

## [Editor Report · Acceptance letter]

2 Jul 2024

PONE-D-23-35794R1 

PLOS ONE

Dear Dr. Lechmann, 

I'm pleased to inform you that your manuscript has been deemed suitable for publication in PLOS ONE. Congratulations! Your manuscript is now being handed over to our production team.

Kind regards, 

on behalf of

Dr. Gianmarco Ferrara 

Academic Editor

PLOS ONE